# Case Study of Additively Manufactured Mountain Bike Stem

**DOI:** 10.3390/ma16134717

**Published:** 2023-06-29

**Authors:** Filip Véle, Michal Ackermann, Jakub Macháček, Jiří Šafka

**Affiliations:** 1Faculty of Mechanical Engineering, Technical University of Liberec, Studentská 1402/2, 461 17 Liberec, Czech Republic; 2The Institute for Nanomaterials, Advanced Technologies and Innovation, Technical University of Liberec, Studentská 1402/2, 461 17 Liberec, Czech Republic; michal.ackermann@tul.cz (M.A.); jakub.machacek@tul.cz (J.M.); jiri.safka@tul.cz (J.Š.)

**Keywords:** selective laser melting, additive manufacturing, bike stem, topology optimisation

## Abstract

This article is focused on a case study of the topology optimisation of a bike stem manufactured by selective laser melting (SLM) additive technology. Topology optimisation was used as a design tool to model a part with less material used for transferring specific loads than the conventional method. For topology optimisation, Siemens NX 12 software was used with loads defined from the ISO 4210-5 standard. Post-processing of the topology-optimised shape was performed in Altair Inspire software. For this case study, the aluminium alloy AlSi10Mg was selected. For qualitative evaluation, the mechanical properties of the chosen alloy were measured on the tensile specimens. The design of the new bike stem was evaluated by Ansys FEA software with static loadings defined by ISO 4210-5. The functionality of the additively manufactured bike stem was confirmed by actual experiments defined by ISO 4210-5. The resulting new design of the bike stem passed both static tests and is 7.9% lighter than that of the reference.

## 1. Introduction

Additive technologies are progressively developing manufacturing processes that are increasingly finding their way into industrial production. Thanks to the development of new technologies and materials, additive manufacturing is moving from prototypes to fully functional components with unique properties [1,2]. The main advantage of additive technologies over conventional machining is the almost unlimited freedom in the design of new parts. Nowadays, most of the design is solved using standard modelling tools, which leads to shapes suitable for conventional manufacturing processes such as milling and turning. However, it is crucial to use different modelling methods for additive technologies. There are various rules for modifying the part to exploit the potential of additive manufacturing [3]. These rules are contained in the term additive design [4,5]. Nowadays, topology optimisation methods are more often used for designing parts that are manufactured by additive technologies [6].

The connection between topology optimisation-based design and additive manufacturing brings exceptional possibilities for creating highly efficient parts and structures. The main advantage of this design approach is that it uses the least possible amount of material to fulfil the part’s purpose. A low amount of material is a very beneficial feature for the AM process because it usually leads to lower build time to manufacture the part. Moreover, the final product is lighter, and its cost may be substantially reduced. When applying topological optimisation, it is necessary to know the exact mechanical properties of the materials, the external loads applied on the part and specific technological constraints related to selected additive technology [7,8]. Because of the growing interest in design for additive technologies, topology optimisation or other mathematical designing methods are implemented into standard CAD software. The main advantage is that the software does not require programming language knowledge, making it more suitable for a wider group of designers [9].

This presented case study is focused on using topology optimisation to design a new mountain bike stem that can be manufactured by selective laser melting technology from AlSi10Mg powder material. In conventional production, this aluminium alloy is used for casting parts with thin walls and very complex geometries. Considering its strength, hardness, and dynamic properties, it is used for products subjected to higher force loads. This alloy is used in aerospace and automotive applications due to its good mechanical properties and low weight [10,11,12,13,14]. The mountain bike stem was selected with regard to combined bending and rotational loads occurring on the part while riding a bicycle. Several researchers have already used mathematical optimisation methods for redesigning bicycle components with regard to additive technologies. Moreno used topology optimisation to design the bike stem from Polyamide 6. The author reached a weight reduction of about 53.8% from the initial design space [15]. For topology optimisation, load cases were calculated from the weight of cyclists. Jost used generative design optimisation for creating the bike stem. The author used titanium alloy Ti64 and reached a weight reduction of about 26% of initial design space [16]. In the mentioned case study, the load cases for generative design optimisation were taken from a study of forces applied to a bicycle during normal cycling [17].

The novelty of this work is twofold. Firstly, all the measures were taken to develop a fully functional AM bike stem with possible commercial applications. In contrast with cited authors, the part was designed to pass demanding tests defined by ISO 4210-5 [18]. Secondly, this work did not end up solely with design but with manufacturing the part and, finally, experimental testing of its structural properties.

## 2. Materials and Methods

This article focuses on designing a new shape of a mountain bike stem with the help of a topology optimisation tool. The bike stem will be additively manufactured on SLM technology from AlSi10Mg powder material. The case study is divided into five steps. The first step is about finding the mechanical properties of parts manufactured under selected process parameters. Then follows the definition of boundary conditions, design space and load cases from ISO 4210-5. The next step is running the topology optimisation and FEA analysis of the optimised results. The following step contains the production of the bike stem and measuring the deviations from CAD data. The last step is the experimental validation of the bike stem under the worst load cases defined by ISO 4210-5.

### 2.1. Selective Laser Melting Technology

According to ISO 52900 [19], SLM is classified as a powder bed fusion–laser beam (PBF-LB) additive technology. This technology uses a laser as an energy source to melt the metal powder. A schematic of the technology is shown in Figure 1.

The principle of the technology is based on the deposition of a thin layer of metal powder on the order of tens of micrometres. The material is spread evenly over the substrate plate, and thus a continuous layer of powder is formed. The cross-section area of the CAD data is scanned with a laser beam. This is followed by a technological pause of a few seconds to stabilise the build surface and filter the internal nitrogen (N${}_{2}$) or argon (Ar) shielding atmosphere from the combustion gases generated during the scanning process. During this pause, the substrate plate decreases by the height of the selected layer. The whole process is repeated until the desired part is finished. In the final stage, the build chamber, powder, and the part are cooled to ambient temperature [20,21]. In today’s market, many types of metal powders can be found from various producers. Moreover, research teams worldwide deal with the development of new materials and testing of their properties [22,23,24].

In comparison with other metal-based additive technologies, the SLM produces fully dense products straight from the process. Specifically, no other thermal or chemical processes are needed to make the material solid. For instance, metal fused filament fabrication needs chemical debinding, followed by metal sintering [25]. A similar process must be applied in the case of Metal Binder Jetting [26]. SLM technology finds its application in prototyping, but due to its properties, it is also used to produce fully functional parts in the automotive, aerospace, and other industrial branches. Even in medicine, the SLM finds its application in producing implants from titanium and Co-Cr alloys [27]. SLM technology is often used to produce injection moulds and machining tools with conformal cooling channels [28,29]. The actual production using SLM technology is demanding in regard to data preparation for printing, maintenance of the printer, and its operation. Therefore, it is important to utilise all the possibilities which this technology offers. Designers should consider the amount of material used for a desired part. This fact leads to the use of topology optimisation and the implementation of lightweight structures.

### 2.2. SLM Process Parameters

In SLM technology, it is possible to influence dozens of process parameters that specifically influence the final quality of the produced part [30]. However, the most influential are the four basic parameters that are contained in the equation of volumetric energy density (VED, Equation (Equation 1)), which is defined by laser power (*P*), hatch distance (*h*), layer thickness (*t*), and scanning speed (*v*). The VED defines the energy that is delivered into the powder base. This value is used for fast comparing the used process parameters in the manufacturing process.
(1)$$VED=\frac{P}{v\;\cdot \;h\;\cdot \;t}$$

In this study, process parameters described in Table 1 were used. The stripe hatch scanning strategy was chosen according to the lowest values of porosity measurement realised on cubes with an edge size of 10 × 10 × 10 mm.

### 2.3. Material and Characterisation

In this study, AlSi10Mg material was chosen. The chemical composition of the powder version of AlSi10Mg from the datasheet of producer SLM Solutions Group AG (Lübeck, Germany) is shown in Table 2.

The mechanical properties of the SLM manufactured part from this material depend on the process parameters, heat treatment, and ultimately, the part orientation on the substrate plate [31,32,33,34,35]. AlSi10Mg alloy has been well studied by research teams worldwide. Many publications dedicated to this alloy include verification of mechanical properties, heat treatment effect, and substrate plate preheating [13,36,37,38]. This makes it possible to get an idea of the parameters used for printing.

Heat treatment (HT) is a controlled process in which the material structure changes and the mechanical properties are altered. These processes allow for changes in ductility, strength, hardness, etc. Heat treatment of AlSi10Mg alloy is often applied to increase ductility at the expense of strength. This can be observed across all publications [31,32,33,34,35]. Stress relief (SR) heat treatment is often chosen for SLM technology because of the stress between the layers during printing. This is due to the laser beam’s rapid energy transfer and the area’s subsequent very rapid cooling [39]. The following steps define the setting of the heat treatment used in this study. Heating to 300 °C for 1 h from ambient temperature and then holding for 3 h at this temperature [40]. This is followed by free cooling in the furnace chamber. A diagram of the progression of this HT is shown in Figure 2.

The experiments necessary to determine the mechanical properties of AlSi10Mg material, including the effect of the heat treatment used, were carried out. To compare the mechanical properties, tensile test specimens were fabricated according to ISO 6892-1 [41] with circular cross sections with a diameter of 6 mm. In this study, it was decided to perform tensile tests on specimens that were not machined. After printing, specimens were ground using 320 grit/cm${}^{2}$ sandpaper. The results from these tests were compared with the machined samples to observe any differences in mechanical properties. The test was carried out according to ISO 6892-1 with a 5 mm/min load rate using an MFL 800B extensometer (MF Mess & Feinwerktechnik GmbH, Velbert, Germany) on a TiraTest universal tensile test device. The specimens were manufactured in horizontal (H) and vertical (V) orientations relative to the substrate plate. The overview of printed specimens and the applied HT, surface machining and orientation is shown in Table 3. Each batch contained five specimens.

### 2.4. Standard for Testing Bicycle Components ISO 4210-5

This standard contributes to the highest possible safety in bicycle operation, and for our purpose, it describes the tests that a bike stem has to pass. Two static tests with the worst possible loads applied on the bike stem defined by ISO 4210-5 were carried out (Figure 3). The tests must be realised in an assembly with a handlebar of the maximum length for which the bike stem can be used. For this study, a Force Basic H4.2 handlebar (KCK Cyklosport-Mode Ltd. Otrokovice, Czech Republic) with a width of 680 mm and a 6° sweep towards the rider was selected.

The first load test is the side bending, shown in Figure 3a. In this test, the bike stem is clamped in a fixture that replaces the head tube. The bike stem is fastened to the head tube through bolts tightened to a torque of 5 Nm. The handlebar is attached using clamps and bolts with the same tightening torque. The load for mountain bikes defined by the ISO standard is applied 50 mm from the edge of the handlebar on one side with a force magnitude F${}_{2}$ = 1000 N. The test is successful if the bike stem can withstand one minute without signs of cracking and displacement of the point where the force is applied is less than 15 mm.

The second load test is the forward bending, shown in Figure 3b. In this test, the alignment of the bike stem and handlebar is the same as in the side bending test. The force is applied to the handlebar clamping section. This test is divided into two levels. In the first level, the loading force is F${}_{3}$ = 1600 N with a duration of one minute, and if there is no destruction or displacement greater than 10 mm, the load is gradually increased to F${}_{4}$ = 2600 N with the same action for one minute. If, after the second phase, there is no destruction or visible cracks, the test is successful.

### 2.5. Topology Optimisation

The topology optimisation method was chosen as a tool for designing a new shape of a bike stem. It is a mathematical approach to the part’s design, optimised by numerical calculations concerning the specified boundary conditions, material properties, and forces applied on the part [42]. For the optimisation setup, it is essential to correctly set the design space, which is the area where the material distribution can be optimised concerning the objective function defined by the user [43]. These objective functions can be set for weight reduction, stiffness increase, etc.

The design that results from topology optimisation usually contains surfaces and shapes that cannot be machined by conventional methods. For this reason, topology optimisation is suitable for powder-based additive manufacturing technologies, where the full potential of this method is exploited [7].

Topology optimisation of the bike stem was performed in the Siemens NX 12 (Siemens Industry Software Inc., Plano, TX, USA) software topology optimisation module. The reference bike stem was chosen with a length of 100 mm and a pitch angle of 6°. Therefore, a design space of 100 × 80 × 80 mm was created, which is shown in Figure 4. In this design space, two cylindrical surfaces were created to represent the seating surfaces for the handlebar and head tube. Those surfaces were offset by 3 mm. This volume was excluded from the design space in the case of removing the elements during topology optimisation.

The next step was to set topology optimisation design boundary conditions. Due to the nature of the bike stem’s symmetric shape, the XY plane of symmetry was created, shown in Figure 4. This plane ensures the topology optimisation will create symmetric results even if the load is not applied symmetrically.

Next, the loading forces were defined. These were taken from the ISO 4210-5 standard for the testing of bike stems. These forces were divided into two load cases. The first state contained a force F${}_{2}$ = 1000 N applied 50 mm from the edge of the handlebar, and the second state was defined by force F${}_{4}$ = 2600 N applied at 45° in the plane of symmetry to the point where the handlebar is attached. Table 4 contains values of mechanical constants used in this case study. The last step in the topology optimisation definition is selecting an objective function. In this regard, the “maximize stiffness” approach was used with target mass constraint. Desired mass was selected according to the market analysis of bike stems. The weight of the commercially available bike stem is from 100 g to 140 g. The lowest value, 100 g was selected as a value for this constraint.

### 2.6. Evaluation of Topology-Optimised Results

After the topology optimisation process, it is important to verify the design through finite element analysis (FEA). The Ansys Workbench 21 (Ansys, Inc. Canonsburg, PA, USA) was used for stress–strain analysis. The model for the FEA simulation was created to be as close as possible to the assembly of the bike stem and handlebar. The mechanical properties of the selected component materials are given in Table 5. The element types and mesh sizes are described in Table 6. The element size was selected according to mesh sensitivity validation.

The FEA evaluation was performed based on the von Mises equivalent stress. The individual tests from the ISO 4210-5 standard contain information on the maximum displacement during force load. Therefore, the total deformation of the stem and handlebar assembly was also evaluated.

## 3. Results

### 3.1. Mechanical Properties of AlSi10Mg

The results from the tensile test are shown in Figure 5 and Figure 6. The worst mechanical properties were obtained on heat-treated specimens printed in a vertical orientation. For this reason, it was decided not to orient the bike stem in a purely vertical direction.

The choice of SR heat treatment was made considering the worst-case scenario, which occurred in vertical orientation without HT. In this case, a ductility of only 1.5% was achieved. The HT was chosen to prevent such a condition from occurring on the actual part. The mechanical properties used for topology optimisation and FEA calculation were derived from the tensile test of printed specimens. The mechanical constants used for simulations were YS = 220 MPa, UTS = 340 MPa, and E = 68 GPa. These values were approximated from heat-treated, horizontal-printed specimens marked as Batch 4 in Figure 5 and Figure 6.

### 3.2. Topology Optimisation of Mountain Bike Stem

The NX 12 topology optimisation module enables the control of material distribution throughout the design space. The value of this command is from 0 to 100%. In the case of 100%, the resulting optimisation creates solid parts. In the case of using 0%, the part is modelled with thin truss structures. After several simulation attempts with various percentage settings, it was decided to set a value of 43%. From a design perspective, this value gives great results in a truss cross-section, which is suitable for the SLM manufacturing process. This setting positively affected the resulting internal stresses and the mass of the simulated bike stem.

The resulting optimisation, shown in Figure 7, took 8 h and 10 min to converge. The optimisation algorithm reached a stem weight of 98 g, thus meeting the target weight requirement of 100 g.

The model, created in the Siemens NX 12 topology optimisation module, contains some imperfections and places where significant stresses have been generated that exceed the strength limit of the material. This is mainly caused by the initial design space shape. For this reason, it was necessary to redesign the shape to a printable state. Altair Inspire 2021.1 (Altair Engineering Inc., Troy, MI, USA) software was used for this step using the Polynurbs function, as shown in Figure 8. The remaining parts of the design used for clamping the handlebar and head tube were redesigned in Geomagic Desing X 2022.0.0 (3D Systems Corporation, Rock Hill, SC, USA) software. The resulting shape of the mountain bike stem is shown in Figure 9.

### 3.3. FEA Analysis of Final Bike Stem Design

The final design was used for FEA analysis in Ansys Mechanical 2021. The simulation model is shown in Figure 10.

The contact between the head tube and the bike stem was defined as “bonded”, and contact between the handlebar and the bike stem was defined as “no separation” in the case of movement during the simulation. The head tube was modelled as a rigid body. The handlebar model complies with Force Basic H4.2 used in the experiment. This body is deformable and clamped to the bike stem by four bolts, each of which was tightened to a torque of 5 Nm.

#### 3.3.1. Side Bending Test

The deformation and stress results of side bending simulation are shown in Figure 11 and Figure 12. The maximal displacement at the point of force application is 14.332 mm. This value has to be less than 15 mm to meet the ISO 4210-5 standard.

The stress values are slightly above the yield strength of the material in several regions in the range of 225 to 251 MPa. The formation of permanent deformation after the experiment can be assumed in the clamping section of the bike stem where the handlebar is connected.

#### 3.3.2. Forward Bending Test

The deformation results of the first level of forward bending simulation with force load F${}_{3}$ = 1600 N are shown in Figure 13.

A maximal displacement of 0.41 mm occurs on the part during simulation. This value has to be less than 10 mm to meet the requirements defined by ISO 4210-5. For the second level with used force F${}_{4}$ = 2600 N the deformation results are shown in Figure 14. Here, a maximal displacement of 0.84 mm occurs on the part during simulation. For the second level, maximal displacement is not defined by the ISO standard. The displacement is still less than 10 mm; therefore, it still fits into the requirements defined by ISO 4210-5 for the first test.

In Figure 15, the results of the first level of stress analysis are shown. The stress values are below the yield strength; hence, no significant permanent deformation is expected to occur on the bike stem after the experiment. In Figure 16, the results of the second level of stress analysis are shown. The stress values are slightly higher than in the first level but still lower than the yield strength. Permanent deformation is not expected even in the second level of the forward bending experiment.

### 3.4. Fabrication

After the design was verified by FEA simulation, four pieces of bike stems were manufactured (Figure 17) with the setting of process parameters defined in Table 1. Bike stems were printed on an SLM 280HL machine from SLM Solutions Group AG (Lübeck, Germany). The printing process of four pieces of bike stems took 30 h. After printing, the parts were scanned to measure the dimensional accuracy. This was followed by heat treatment, removal of support structures and machining of the seating surfaces.

After the machining of the seating surfaces, the weight measurement of the bike stem was carried out as described in Table 7. The table shows an average saving of about 7.9% on the body of the bike stem compared to a conventionally manufactured bike stem and 27.87% weight saving on clamps. As the reference, Force Team Spry (KCK Cyklosport-Mode Ltd. Otrokovice, Czech Republic) bike stem was selected with 100 mm length and 6° slope. For weight measurement, RADWAG WTC 600 digital scale (Radwag, Radom, Poland) was used.

### 3.5. Experimental Testing of Printed Bike Stems

Experimental load tests according to ISO 4210-5 were performed on a test bench as a bike stem and handlebar assembly. The first experiment was a side bend, which is shown in Figure 18a. The bike stem assembly was attached to a rigid fixture on the head tube seating surface and tightened with M5 bolts by a torque of 5 nm. The force required by the ISO 4210-5 standard was applied through a pneumatic cylinder. The displacement of the handlebar was measured at the point where the force was applied using MarCator 1086R digital indicator (Mahr GmbH, Göttingen, Germany). The resulting measured displacement was 14.72 mm, and it fit into a limit defined by ISO 4210-5 standard.

The second experiment was forward bending with a force magnitude of F${}_{3}$ = 1600 N. The bike stem position on the testing bench is shown in Figure 18b. The method of force load and measurement of displacement was realised, similar to the previous experiment. The resulting measured displacement was 0.50 mm and fit into a limit defined by ISO 4210-5. The second level with force magnitude F${}_{4}$ = 2600 N generates displacement of 0.92 mm, which again fits into the ISO standard limitation.

### 3.6. Measuring the Deformation on Parts

The dimension accuracy analysis was performed on bike stems at each stage of the production process. Two bike stems were used for this shape accuracy analysis. One was left attached to the substrate plate (Version A), and the second was separated from the substrate plate right after the manufacturing process (Version B). Furthermore, two more bike stems were measured after the experiments to map the permanent deformation. For scanning, the optical scanner Range Vision Spectrum (RangeVision CIS, Moscow, Russia) was used. For better accuracy of scanned data, reference points and a rotation table were used.

Gom Inspect 2020 (Carl Zeiss AG, Oberkochen, Germany) was used to validate overall deformation. The alignment of scan data on the STEP model was performed by a global best-fit function. On the body of the stem, twelve evaluation points were selected for the comparison of deformation between individual production stages (Figure 19). These points are located in such areas where no support structures were attached. This ensures that measurements are not influenced by any finishing operations.

After the manufacturing process, the first bike stem (A1) was scanned on the substrate plate. The bike stem was aligned on step data with a deviation of 0.07 mm. After alignment, dimension accuracy analysis was performed (see Figure 20).

The summarised data from five measurements with different settings are contained in Table 8. The measurement after the manufacturing process without any HT shows a maximal deviation of 0.15 mm. The higher deformation occurs in the clamping section of the handlebar and head tube. For this stage, normal deviations in the area of the truss section are in the range of ±0.10 mm.

For the second measurement, the bike stem (A2) was still left on the substrate plate, and the HT was performed. The alignment of scanned data on the STEP model was with a deviation of 0.08 mm. The results of the analysis are shown in Figure 21. The results of measured points are nearly the same in the truss and head tube section. The only higher deviation of 0.21 mm can be found in the handlebar clamping section.

The last measurement of the first bike stem (A3) was after removal from the substrate plate. The results of the analysis are shown in Figure 22. In this case, the alignment deviation was 0.075 mm, and the measured data are with little deviation from the previous measurements. The maximal deviation still occurs in the handlebar clamping area.

After the manufacturing process, the bike stem (B1) was subtracted from the substrate plate, and the support structures were removed. The bike stem was aligned with the same setting as in previous measurements with a deviation of 0.10 mm. The results from the analysis are shown in Figure 23, where the maximal deviation is 0.23 mm occurs in the handlebar clamping area.

The truss section also has a higher deviation than what was measured on the bike stem that was left on the substrate plate (A1).

After HT the bike stem (B2) was measured again with a data deviation of 0.07 mm. The results of the analysis are shown in Figure 24. The maximal deviation is 0.28 mm in the handlebar clamping area.

From all realised measurements is seen a worsening trend of bike stem deviation from nominal data when the bike stem is removed from the substrate plate before HT. It is suitable to leave parts on the substrate plate during HT, and it leads to better dimensional accuracy than if the bike stem is removed before HT.

In the last step, the bike stems after experiments were analysed. It was necessary to evaluate the permanent deformation on the part. The first evaluated bike stem was after the side bending experiment. The scanned mesh was aligned on STEP data with a deviation of 0.14 mm. The analysis of permanent deformation is shown in Figure 25 where the maximal deviation in the handlebar clamping area is 0.61 mm. Also, the truss section was influenced by applied force, and the deviation is around 0.3 mm. This permanent deformation was produced when the yield strength of the material was exceeded during the experiment.

The scanned data of the bike stem after forward bending were aligned with a deviation of 0.13 mm. The results shown in Figure 26 correspond to the second level of forward bending with force F${}_{4}$ = 2600 N. The maximal deviation that occurs on the bike stem is 0.47 mm and is located in the handlebar clamping area.

The outcomes from the analysis show that even after passing these two experiments, the bike stem is suitable for further use.

## 4. Discussion

In the first phase of this study, the mechanical properties of the AlSi10Mg aluminium alloy were tested. The tensile tests were realised with and without HT in horizontal and vertical orientations relative to the substrate plate. Results show that using heat treatment, YS and UTS are reduced at the expense of increased ductility. Various authors report this phenomenon [31,32,33,45]. Our results also show approximately a 15% difference between these stress levels for vertically and horizontally manufactured specimens. Usually, authors who deal with the same topic do not observe such a high anisotropy. One reason might be that only the non-machined specimens were tested in the vertical direction. Some surface imperfections might have occurred on the borders of the parts, causing lower strength in vertically oriented specimens. This outcome needs further study to fully understand it.

Reduced ductility in as-built conditions was observed for specimens manufactured in the vertical direction. After the heat treatment was applied, yield strength and ultimate tensile strength decreased significantly. Thus, it was decided not to orient the final product in a vertical position. The mechanical constants for vertical orientation with applied HT reached YS = 170 MPa and UTS = 300 MPa. Those constants correspond with results from Rosenthal’s [46] study, where YS = 183 MPa and UTS = 286 MPa were measured.

For topology optimisation and FEA analysis, mechanical constants YS = 220 MPa, UTS = 340 MPa, and E = 68 GPa were used. These values correspond to horizontally manufactured and heat-treated specimens. The most similar process settings were used in the studies of Van et al. [47] and Patakham et al. [48], where the YS = 210 MPa, UTS = 325 MPa, and YS = 216 MPa, UTS = 332 MPa were reported, respectively.

The main phase of this study was to create a new design of a mountain bike stem using topology optimisation. The definition of applied forces on the bike stem was performed according to the ISO 4210-5 standard instead of calculating them from the rider’s weight, as was performed in the study [15]. The ISO standard is stricter for introducing the safety level into a design. According to market analysis, the required weight of 100 g of the final bike stem was selected as a mass constraint for the topology optimisation algorithm. After topology optimisation and redesigning the shape, the final bike stem was about 7.9% lighter than that of the reference. It is a very good improvement because a material with similar properties to the reference bike stem was used instead of employing such material as Ti64V, which has higher strength.

The FEA simulation of the topology-optimised bike stem was performed in Ansys Workbench software. The displacement of the handlebar during a side bending experiment was 14.332 mm, which fits into a limitation defined in the ISO standard. After the real experiment, the measured displacement was 14.72 mm, the ISO standard limit was not reached, and the experiment was successful. With regard to exceeding the yield strength during the FEA simulations, the formation of permanent deformations on the bike stem after the performed experiment was assumed. This was confirmed by dimension accuracy measurement. The exceeded yield strength affected the truss and handlebar clamping section the most. Despite forming permanent deformation, the structure of the bike stem remained without any cracks or other critical failures.

The second experiment was forward bending. The simulation shows a displacement of only 0.41 mm, which is far from the limitation of the ISO standard. The stress values are lower than the yield strength of the material, hence the permanent deformation of the bike stem was not expected. After the first level of the experiment, the displacement of only 0.50 mm was measured. During the second level of the experiment, the displacement rose to 0.92 mm, which still fits into limitations defined by the ISO standard. During the measurement of dimension accuracy, the permanent deformation occurred on seating surfaces, but the truss section was without critical permanent deformation. Comparing the results from both experiments, the topology optimisation creates a design that is much stiffer for forward bending than the side bending load case.

The last part of this case study contains the measurement of the dimension accuracy of the bike stem after the printing process. The effect of leaving or removing the part from the building platform before and after heat treatment was investigated during this part. To obtain the best accuracy of parts, it is better to leave the part on the substrate plate during the HT. When the part is removed from the substrate plate before heat treatment is applied, it is no longer held in place by support structures. Inner stresses are then released, which causes deformation and, thus, possible devaluation of the part.

## 5. Conclusions

This study verified topological optimisation for creating a new mountain bike stem design using SLM technology and AlSi10Mg aluminium alloy. Furthermore, the printing process parameters were verified, including determining the mechanical properties required for the simulation software. Verification of the functionality of the design by static tests defined by ISO 4210-5 was successful in both load cases. In the study, the newly designed bike stem was found to be very stiff in the forward bending test and exhibited a displacement of only 0.5 mm. The resulting topologically optimised mountain bike stem was 7.9% lighter than the commercially produced reference bike stem. From a manufacturing perspective, the need of running the HT prior to part removal from the substrate plate was confirmed. Using this way, lower deformation occurs, and better dimension accuracy of the part is achieved.

The limitation of our work is that only one type of material was studied. The usage of more advanced materials such as titanium alloy may lead to the creation of even lighter and more durable components. However, one must also consider the final cost of the product, which will surely be much higher than for AlSi10Mg alloy. This topic may be interesting for other authors to create a deeper study.

In our future work, this study will be further developed to meet the dynamic tests defined in the ISO 4210-5 standard. In addition, the design and optimisation settings will be modified to reduce the displacement in the case of side bending, which is close to the experimental compliance limit for the current design and handlebar used.

## Figures and Tables

**Figure 1 materials-16-04717-f001:**
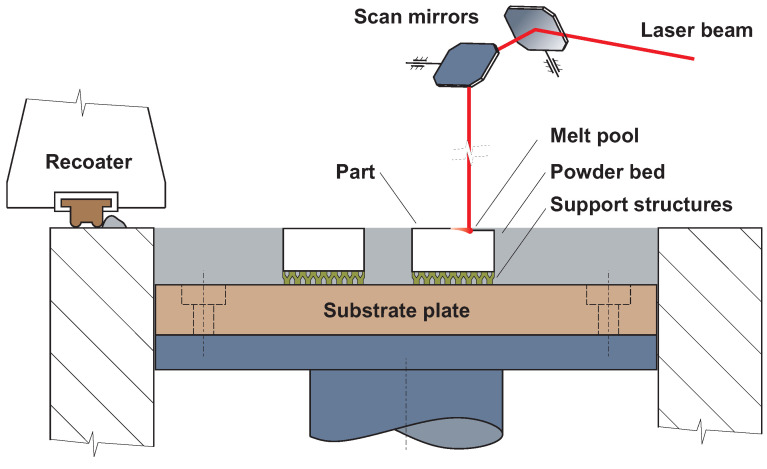
Selective laser melting principle.

**Figure 2 materials-16-04717-f002:**
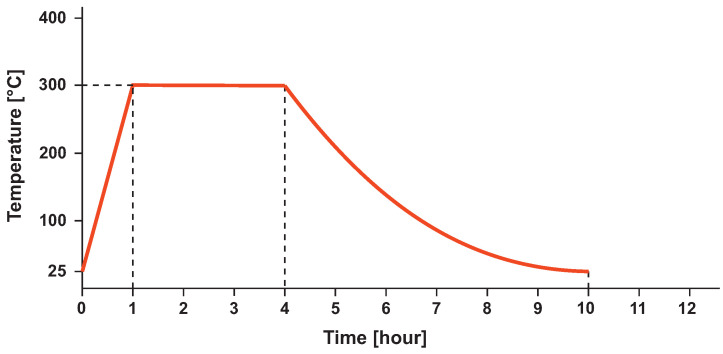
Stress relief heat treatment ramp of AlSi10Mg alloy.

**Figure 3 materials-16-04717-f003:**
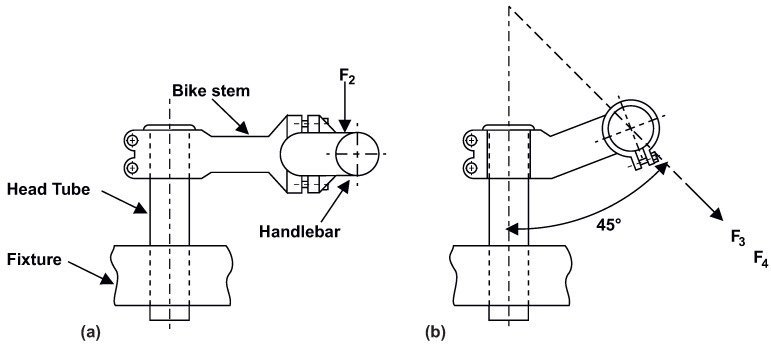
Experiment setting defined by ISO 4210-5 [18].

**Figure 4 materials-16-04717-f004:**
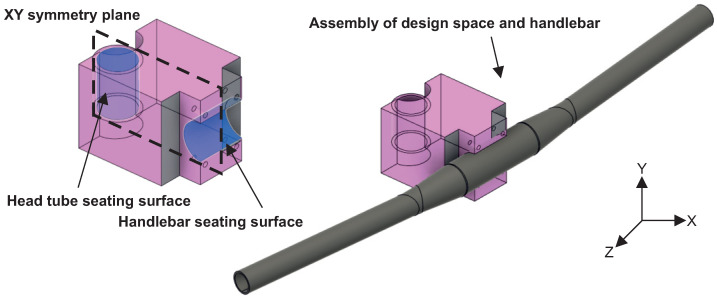
Design space for topology optimisation.

**Figure 5 materials-16-04717-f005:**
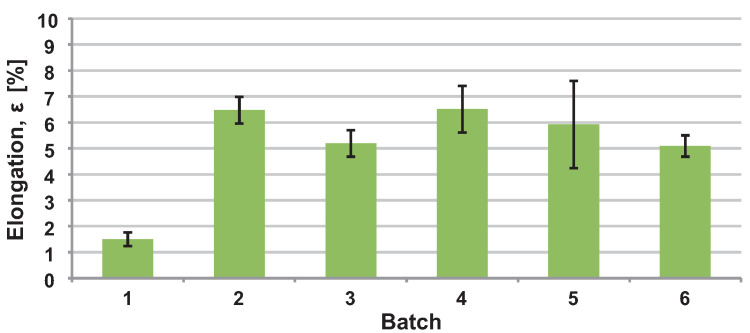
Elongation of printed specimens.

**Figure 6 materials-16-04717-f006:**
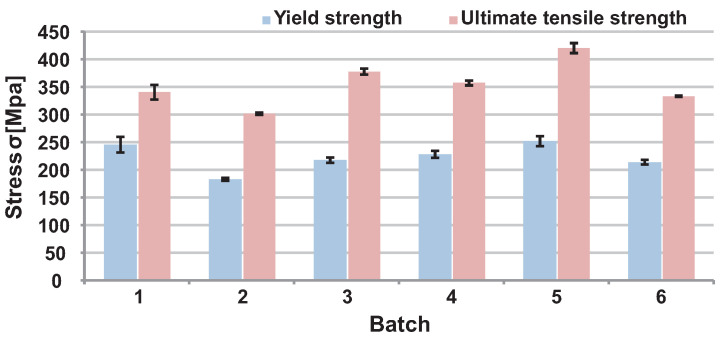
Mechanical properties of printed specimens.

**Figure 7 materials-16-04717-f007:**
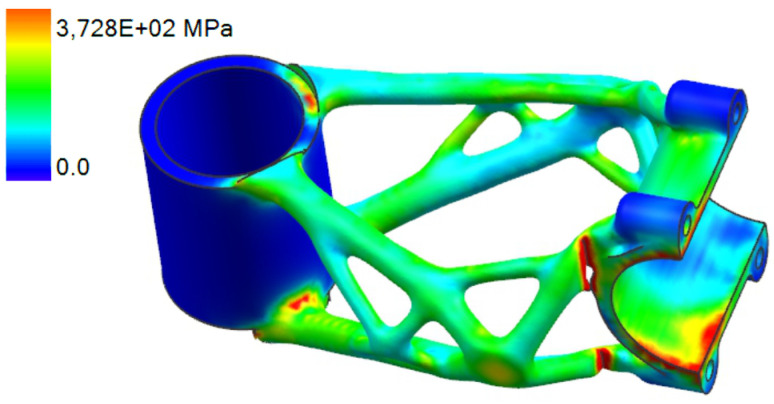
Topological optimised design from Siemens NX 12 software.

**Figure 8 materials-16-04717-f008:**
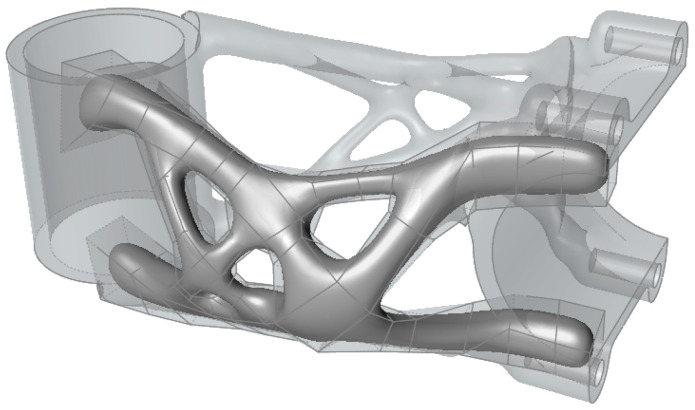
Polynurbs fitting in software Altair Inspire.

**Figure 9 materials-16-04717-f009:**
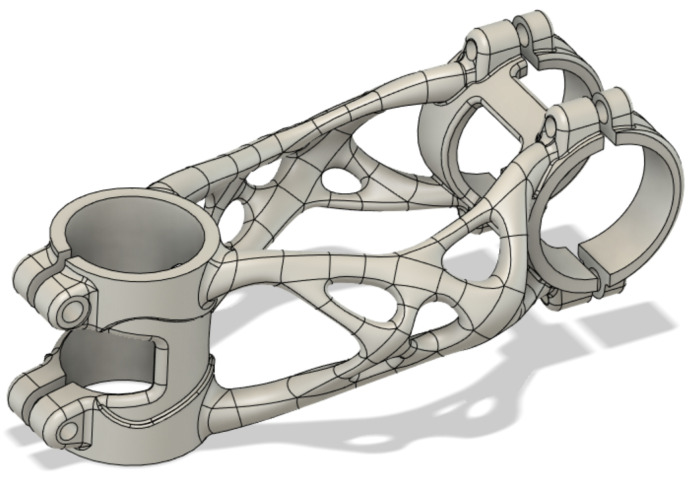
Final design of topological optimised mountain bike stem.

**Figure 10 materials-16-04717-f010:**
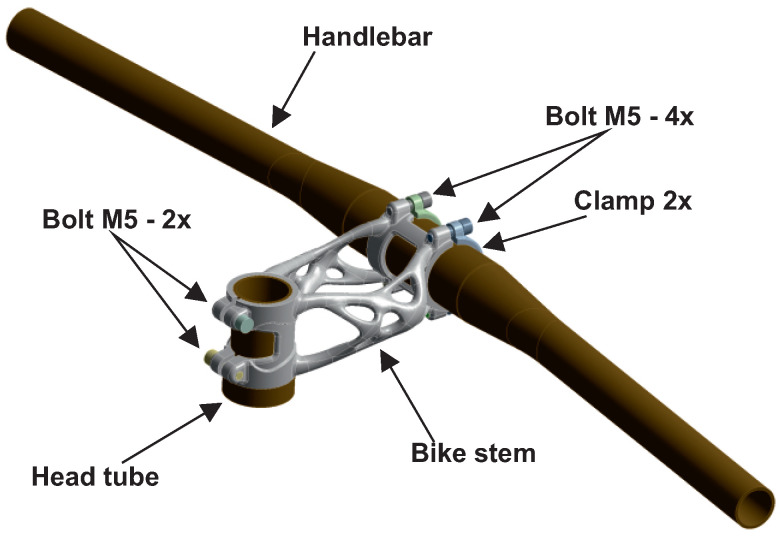
FEA model of bike stem and handlebar assembly.

**Figure 11 materials-16-04717-f011:**
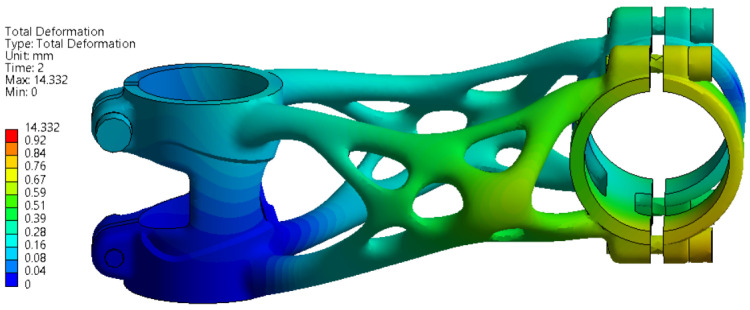
FEA simulation of deformation during side bending experiment.

**Figure 12 materials-16-04717-f012:**
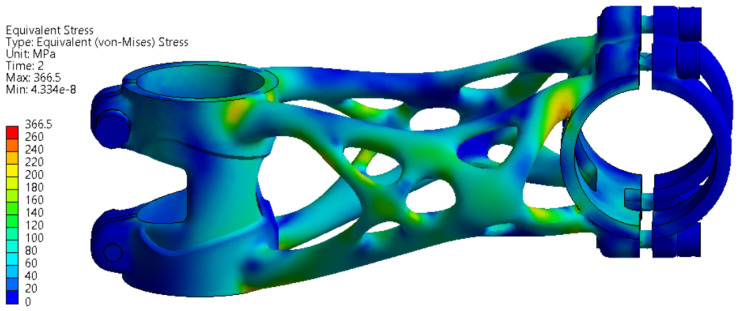
FEA simulation of stress distribution during side bending experiment.

**Figure 13 materials-16-04717-f013:**
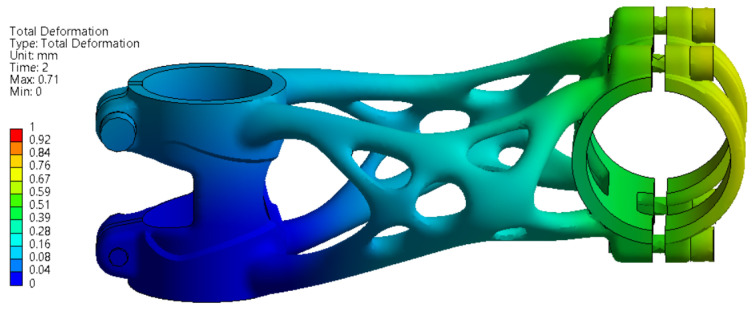
FEA simulation of deformation during the first level of forward bending experiment.

**Figure 14 materials-16-04717-f014:**
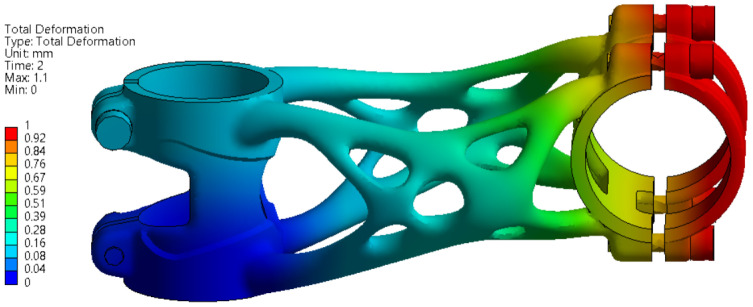
FEA simulation of deformation during the second level of forward bending experiment.

**Figure 15 materials-16-04717-f015:**
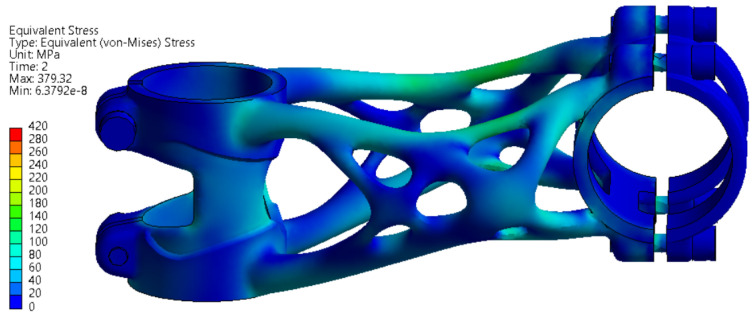
FEA simulation of stress during the first level of forward bending experiment.

**Figure 16 materials-16-04717-f016:**
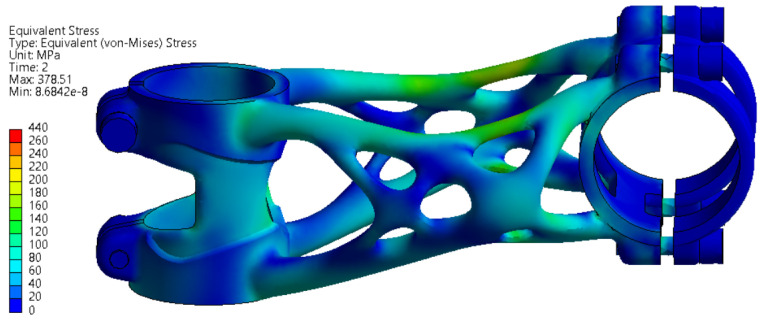
FEA simulation of stress during the second level of forward bending experiment.

**Figure 17 materials-16-04717-f017:**
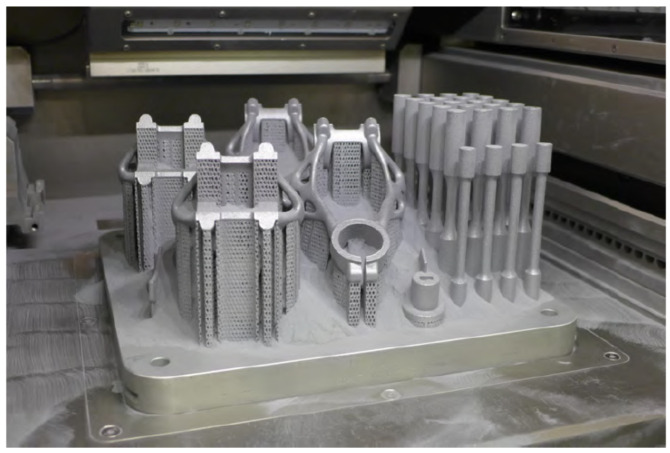
Manufactured bike stems in SLM build chamber.

**Figure 18 materials-16-04717-f018:**
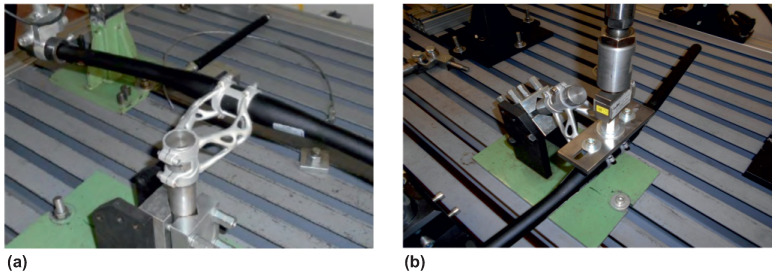
Test bench for experiments: (**a**) side bending (**b**) forward bending.

**Figure 19 materials-16-04717-f019:**
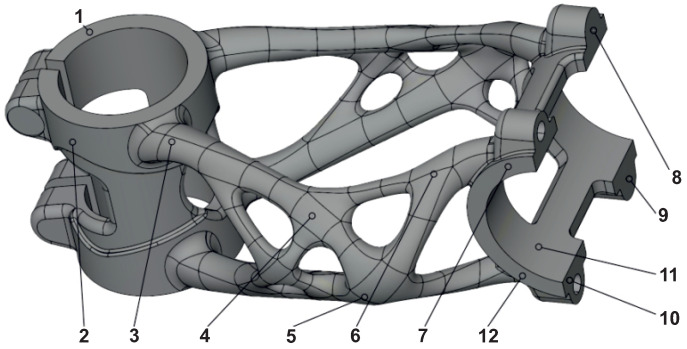
Selected points for deformation measurement.

**Figure 20 materials-16-04717-f020:**
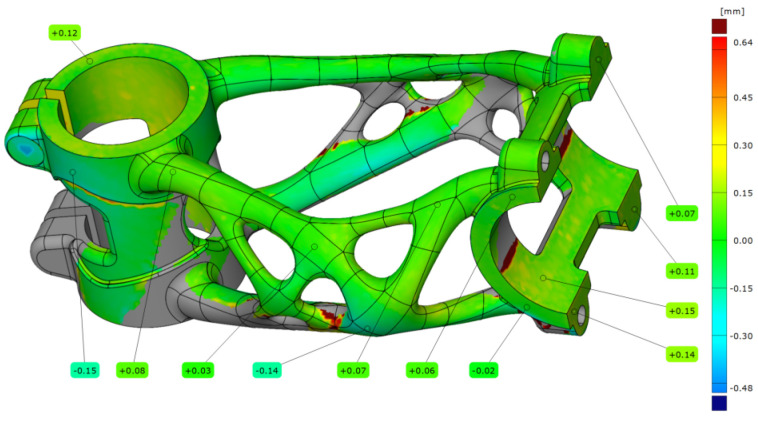
Deviation map of bike stem (A1).

**Figure 21 materials-16-04717-f021:**
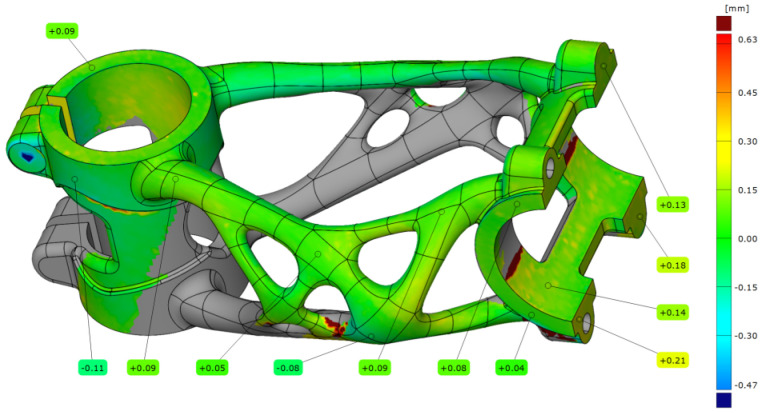
Deviation map of bike stem (A2).

**Figure 22 materials-16-04717-f022:**
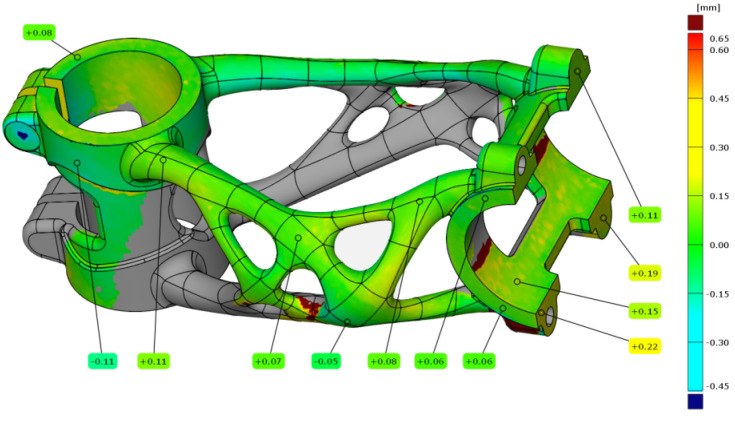
Deviation map of HT bike stem (A3).

**Figure 23 materials-16-04717-f023:**
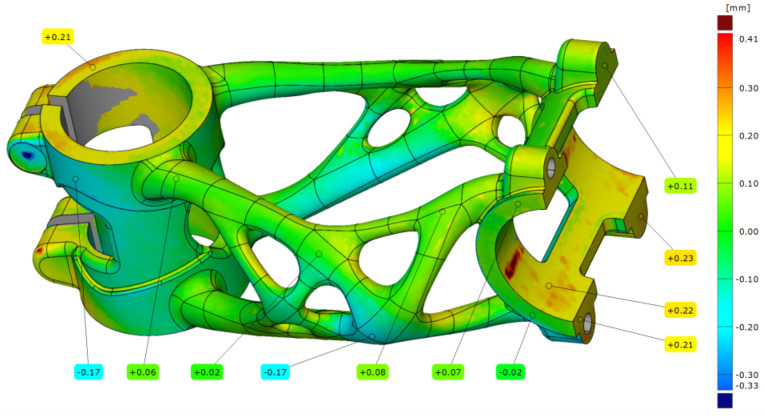
Deviation map of bike stem (B1).

**Figure 24 materials-16-04717-f024:**
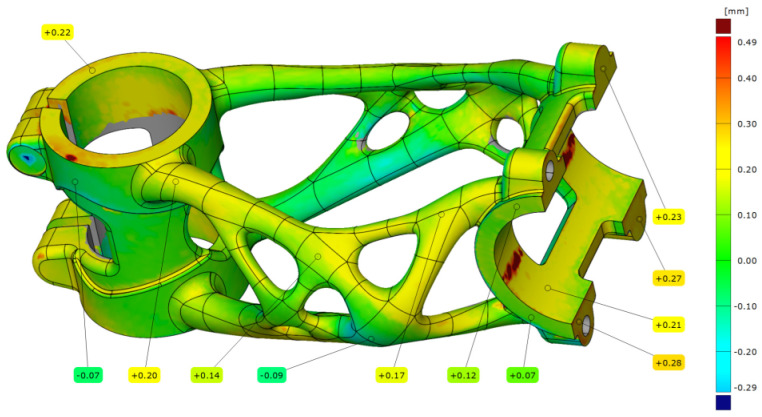
Deviation map of HT bike stem (B2).

**Figure 25 materials-16-04717-f025:**
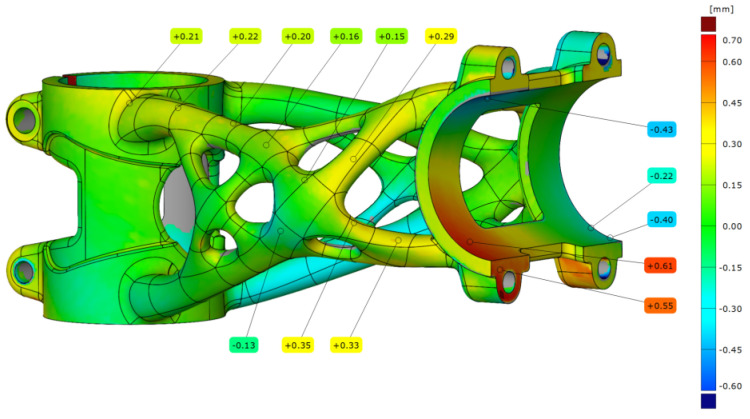
Deviation map of bike stem after side bending experiment.

**Figure 26 materials-16-04717-f026:**
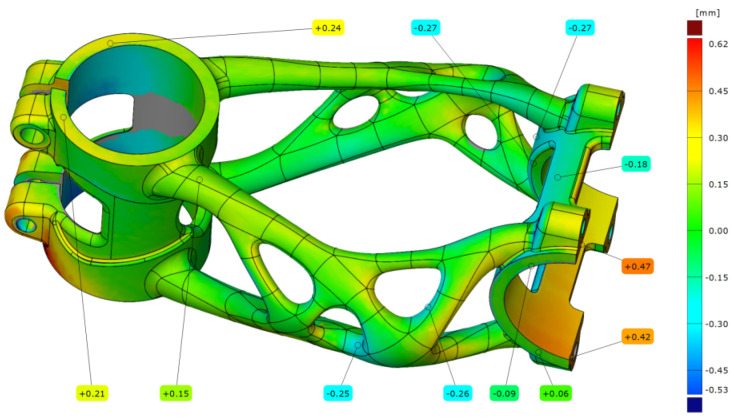
Deviation map of bike stem after forward bending experiment.

**Table 1 materials-16-04717-t001:** Process parameters of AlSi10Mg powder.

Parameter	Value
Laser power (W)	350
Scanning speed (mm/s)	980
Hatch distance (mm)	0.12
Layer thickness (mm)	0.05
Platform preheating (°C)	110
Scanning strategy (-)	Stripes
VED (J/mm${}^{3}$)	59.52
Homogeneity (%)	98.6

**Table 2 materials-16-04717-t002:** Chemical composition of AlSi10Mg powder.

Element	Composition (wt%)	Element	Composition (wt%)
Al	Balance	Zn	0.10
Si	9.00–11.00	Ti	0.15
Fe	0.55	Ni	0.05
Cu	0.05	Pb	0.05
Mn	0.45	Sn	0.05
Mg	0.20–0.45	Rest	0.15

**Table 3 materials-16-04717-t003:** Specimen overview.

Batch	1	2	3	4	5	6
Orientation	V	V	H	H	H	H
Heat treatment	-	SR	-	SR	-	SR
Surface machining	-	-	-	-	Yes	Yes

**Table 4 materials-16-04717-t004:** Mechanical properties of materials used for topology optimisation.

Material	YS (MPa)	UTS (MPa)	E (GPa)	ν (-)
AlSi10Mg	220	340	68	0.33
Al6061 [44]	280	310	68.9	0.33

**Table 5 materials-16-04717-t005:** Mechanical properties of materials used for FEA.

Part	Material	YS (MPa)	UTS (MPa)	E (GPa)	ν (-)
Bike stem	AlSi10Mg	220	340	68	0.33
Clamp	AlSi10Mg	220	340	68	0.33
Handlebar	Al6061 [44]	280	310	69	0.33
Bolts	Steel 8.8	640	800	200	0.30

**Table 6 materials-16-04717-t006:** Element used for FEA.

Part	Element Size	Element Type
Bike stem	0.5 mm	tetrahedron, 4-noded
Clamp	0.5 mm	tetrahedron, 4-noded
Handlebar	1 mm	tetrahedron, 4-noded
Bolts	1 mm	tetrahedron, 4-noded

**Table 7 materials-16-04717-t007:** Weight measurement of bike stems.

Stem Number	Body (g)	Clamp (g)	Bolts (g)	Assembly (g)
Stem 1	89.6	10.1	22.8	122.5
Stem 2	89.3	10.2	22.8	122.3
Stem 3	89.0	10.3	22.8	122.1
Stem 4	88.3	10.1	22.8	121.2
Reference	96.7	14.1	22.8	133.6

**Table 8 materials-16-04717-t008:** Deviation of measured points.

Stem Label	Measured Point (mm)					
	1	2	3	4	5	6
A1	0.12	−0.15	0.08	0.03	−0.14	0.07
A2	0.09	−0.11	0.09	0.05	−0.08	0.09
A3	0.08	−0.11	0.11	0.07	−0.05	0.08
B1	0.21	−0.17	0.06	0.02	−0.17	0.08
B2	0.22	−0.07	0.20	0.14	−0.09	0.17
	7	8	9	10	11	12
A1	0.06	0.07	0.11	0.14	0.15	−0.02
A2	0.08	0.13	0.18	0.21	0.14	0.04
A3	0.06	0.11	0.19	0.22	0.15	0.06
B1	0.07	0.11	0.23	0.21	0.22	−0.02
B2	0.12	0.23	0.27	0.28	0.21	0.07

## Data Availability

The data presented in this study are unavailable due to privacy restrictions.

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
