# Peer review of "Case Study of Additively Manufactured Mountain Bike Stem"

_materials, 2023, doi:10.3390/ma16134717_

Round 1
Reviewer 1 Report
The manuscript is well written and the work carried out has commercial application. However, the authors are requested to address the queries provided below.
· One of the referred works used Polyamide 6 as the material to build the bike stem. In comparison to it, AlSi10Mg is a costly material. Is it feasible and justifiable to use metal additive manufactured part, when cheaper polymer can be used?
· The authors should clearly state how their work is different than the worked reported in the literature review which deals with the bike stem.
· “Worst mechanical properties were obtained on heat-treated specimens printed in a vertical orientation” Why? Any particular reason?
· Also elongation in the case of heat treated vertical printed part is better untreated vertical part. Justify?
One round of proofreading is suggested
Author Response
Thank you for your review. For detailed information, please see attached document.

Reviewer 2 Report
Dear Authors,
the manuscript entitled "Case Study of Additively Manufactured Mountain Bike System" by Filip Vele and co-authors deals with the use of simulation research as well as Selective Laser Melting technology to produce a new shape of bike stem. The paper is well justified, planned and written, and adds to the designing and manufacturing process some new informations. In my opinion, the methodology and results of this study are clear. Conclusions are consistent with the findings. I appreciate the contribution that the Authors made in simulation studies and experimental testing of printed parts as well as preparing the manuscript. However, in my opinion the manuscript needs to be improved in some fields and some general remarks as well as the specific comments are bellow.
Evaluation of the paper, general remarks, editorial comments/typos:
- The title of the article should not contain capital letters alone. The title should be written: "Case Study of Additively Manufactured Mountain Bike System.".
- line 13 - after keywords: Additive manufacruring; and Bike stem; the space is missing.
- line 26 - there is: "additive technologies.[6]" and should be additive technologies [6].
- line 36-38 - the Authors present the goal of the research. Please indicate in this section of the article what its novelty is in relation to the current state of knowledge.
- In line 42 and 45, the Authors use the personal form ("… He reached a weight...."). This is not correct in high-quality articles. It suggests modifying this part of the article. Please check the entire article in terms of personal form.
- In the Introduction chapter, a broader literature review of Selective laser melting technology from AlSi10Mg powder material should be done. Please add more research results description.
- Figure 1 caption - Figure and table captions are not made according to the journal format. The dot is missing at the end of the Figures and Tables caption. Please check entire manuscript.
- line 76 - there is "Industrial brunches", there should be Industrial branches.
- line 128 - there is 5mm, and should be 5 mm.
- Figure 3 caption - there is "...by ISO 4210-5[27]" and should be by ISO 4210-5 [27].
- line 174 - there is 3mm and should be 3 mm.
- Authors present their results with discussion supported by the few research articles. When the results are not discussed and conveniently supported by the open literature, questionable conclusions are obtained. Currently, the article in some part of Discussion chapter looks more like a report from test and simulations than a scientific article. Improvement in the description of the test results is required.
- The above modifications should be implemented before considering the manuscript for publication. I hope these suggestions can help to improve the quality of this paper.
I wish you all the best.
Author Response

(The authors gave the same response as above.)

Reviewer 3 Report
In this paper, the topological optimization of the bicycle rod made by additive selective laser melting (SLM) is studied, and the related process parameters are verified. Research is meaningful, but the following questions need to be answered:
1) In line 27, the author can introduce more about what topological optimization is and what its more specific role and significance is in additive manufacturing.
2) In section 2.1, a description of the advantages of SLM technology compared to other additive technologies and its own characteristics can be appropriately added.
3) As for the experimental parameters in Table 1, how did the author verify the impact of these parameters on part quality?
4) In Section 2.5, what software and algorithms are used in the design of topology optimization? How are design Spaces and loads defined? May be supplemented as appropriate.
5) For the conclusion part of Chapter 5, what are the meanings and limitations of the conclusion? For example, it shows the possibility of innovative design and only discusses a specific material, and further considers more influencing factors. May be supplemented as appropriate.
Author Response

(The authors gave the same response as above.)

Reviewer 4 Report
The manuscript entitled “materials-2470446” dealing with AM bike stem has been reviewed. The paper has been nicely written but needs significant improvement. Please follow my comments.
1. Use laser-based powder bed fusion instead of SLM. Follow ASTM 52900.
2. Figure 1 is very basic. You don’t need to provide such basic information in the text. Please consider removing it.
3. What is the main issue that will be solved by this investigation? Please clarify it in the text.
4. Please add a brief statement on your methodology.
5. Please proofread the paper.
6. How did you select your process parameters. Please add a sentence about this.
7. AM has many usages in different industries. To highlight your work, add a short note in the introduction by using the following papers and mention the privilege of lasers in manufacturing. “Benchmark models for conduction and keyhole modes in laser-based powder bed fusion of Inconel 718”. “Comparative study on the properties of 17-4 PH stainless steel parts made by metal fused filament fabrication process and atomic diffusion additive”.
Proofread the paper.
Author Response

(The authors gave the same response as above.)
